# Doppler lidar at Observatoire de Haute Provence for wind profiling up to 75 km altitude: performance evaluation and observations

Sergey M. Khaykin[1], Alain Hauchecorne[1], Robin Wing[1], Philippe Keckhut[1], Sophie Godin-Beekmann[1], Jacques Porteneuve[1], Jean-Francois Mariscal[1], Jerome Schmitt[2]

1 LATMOS/IPSL, UVSQ, Sorbonne Université, CNRS, Guyancourt, France
2 Observatoire de Haute-Provence ,Université d'Aix-Marseille, CNRS, Saint-Michel l'Observatoire,  France

*Correspondence to*: Sergey Khaykin (sergey.khaykin@latmos.ipsl.fr)

**Abstract.** A direct-detection Rayleigh-Mie Doppler lidar for measuring horizontal wind speed in the middle atmosphere (10 to 50 km altitude) has been deployed at Observatoire de Haute Provence (OHP) in southern France since 1993. After a recent upgrade, the instrument gained the capacity of wind profiling between 5 and 75 km altitude with vertical resolution up to 115 m and temporal resolution up to 5 minutes. The lidar comprises a monomode Nd:Yag laser emitting at 532 nm, three telescope assemblies, and a double-edge Fabry-Perot interferometer for detection of the Doppler shift in the backscattered light. In this article, we describe the instrument design, recap retrieval methodology and provide an updated error estimate for horizontal wind. The evaluation of the wind lidar performance is done using a series of twelve time-coordinated radiosoundings conducted at OHP. A point-by-point intercomparison shows a remarkably small average bias of 0.1 m/s between the lidar and the radiosonde wind profiles with a standard deviation of 2.3 m/s. We report examples of a weekly and an hourly observation series, reflecting various dynamical events in the middle atmosphere, such as a Sudden Stratospheric Warming event in January 2019 and an occurrence of a stationary gravity wave, generated by the flow over the Alps. A qualitative comparison between the wind profiles from the lidar and the ECMWF Integrated Forecast System is also discussed. Finally, we present an example of early validation of the ESA Aeolus space-borne wind lidar using its ground-based predecessor.

## 1 Introduction

Vertically-resolved measurements of the wind velocity in the middle atmosphere are essential for understanding the global circulation driven by dynamical processes such as gravity and planetary waves interacting with the atmospheric flow (Holton 1983). While weather balloon soundings provide regular observations of horizontal wind profiles up to about 30 km altitude, the region of upper stratosphere and lower mesosphere (USLM, ~30 - 75 km) is poorly covered by observations. The only information on the wind field in this layer available on the regular basis is inferred from horizontal pressure gradients derived from space-borne temperature measurements using geostrophic balance assumption (e.g. Oberheide et al., 2002), however this does not allow characterizing regional-scale dynamical processes.

Of particular challenge are the wind measurements in the so-called radar gap between 20 – 60 km (Baumgarten, 2010). Until the early 1990s, the only source of wind measurements in USLM region were the rocket soundings, on which the middle atmosphere wind field climatology was based (Schmidlin, 1986). The high cost of rocket operations has fostered development of remote sensing techniques for wind profiling of the middle atmosphere. Pioneering work in the remote sensing of wind profiles up to the stratopause was conducted by Chanin et al. (1989) at Observatoire de Haute Provence (OHP, 43.9°N, 5.7°E) using incoherent Doppler Rayleigh lidar. Since then, several methods for ground-based lidar measurements of wind using molecular backscattering have been proposed and demonstrated (Bills et al., 1991; Abreu et al., 1992; Tepley et al., 1994; Rees et al., 1996; Friedman et al., 1997; Gentry et al., 2000; Yan et al., 2017). The direct-detection technique for wind profiling has been successfully realized in an airborne Doppler lidar – A2D, Aeolus Airborne Demonstrator (Reitebuch et al., 2009). A2D instrument served a prototype for the most ambitions endeavour in the context of lidar wind profiling – the first ever satellite-borne Doppler lidar instrument ALADIN (Atmospheric Laser Doppler INstrument) (ESA, 2008; Stoffelen et al., 2005), that has been successfully launched by European Space Agency (ESA) in August 2018 (Kanitz et al., 2019).

While the necessity of high resolution (<1 km) wind profiling of USLM region is well recognized (e.g. Meriwether and Gerrard, 2004; Dörnbrack et al., 2017), presently very few instruments with such capacity are operated on a regular or quasi-regular basis. These are Doppler lidar at ALOMAR observatory in northern Norway (Baumgarten, 2010; Hildebrand et al., 2017), LiWind lidar at high-altitude Maido observatory at La Reunion island (Baray et al., 2014) and LIOvent lidar at Observatoire de Haute Provence (OHP), where the pioneering lidar measurements of wind up to 50 km altitude were conducted by Chanin et al. (1989).

The OHP wind lidar was originally designed to cover the height range of 25 – 50 km (Garnier and Chanin, 1992), i.e. where the contribution of Mie scattering by aerosol particles can be neglected in most cases. After the eruption of Pinatubo volcano in 1991 polluting the stratosphere with aerosol up to 35 km, the OHP wind lidar was redesigned to minimize the effect of Mie scattering (Souprayen et al., 1999a,b). The new Rayleigh-Mie Doppler lidar named LIOvent was deployed at OHP in late 1993 and was operated on a regular basis during 1995 – 1999. The observations were used for retrieval of gravity wave parameters and stratospheric wind climatology at OHP (Souprayen et al., 1999a; Hertzog et al., 2001) as well as for a study of the effect of gravity waves on ozone fluctuation in the lower stratosphere (Gibson-Wilde et al., 1997).

After a long period of sporadic operation and limited maintenance, the upgrade of OHP wind lidar was started in 2012. At the same time, a similar wind lidar instrument was deployed at Maido observatory at La Reunion island and passed a thorough performance evaluation, which will be presented in a separate paper. The upgrade of both wind lidars included replacement of the laser, optical filtering elements, and detectors. These efforts were largely motivated by the ESA Aeolus satellite mission carrying ALADIN instrument, which exploits the same principle of Doppler shift detection, i.e. double-edge Fabry-Perot interferometer (Stoffelen et al., 2005; Reitebuch, 2012).

This study aims at characterizing the performance and capacities of OHP LIOvent Doppler lidar after its upgrade. Chapter 2 describes the instrument design; Chapter 3 reports the results of LIOvent validation using 12 collocated radiosoundings;

Chapter 4 provides examples of weekly and hourly observation series; Chapter 5 presents an early case of Aeolus validation using OHP lidar; Chapter 6 concludes the study and sketches the outlook.

## 2 Instrument design and measurement principle

A detailed description of the OHP Doppler lidar (LIOvent) and the methodology for retrieving wind profiles was provided by Souprayen et al. (1999a). Here, we recap the general design of the instrument and its sub-systems after its upgrade, the measurement principle, the error budget as well as the principal stages of data retrieval.

### 2.1 Instrument design

The lidar instrument senses the horizontal wind velocity by measuring the Doppler shift between the emitted and backscattered light of the laser. The Doppler shift corresponds to the projection of the horizontal wind components onto the line-of-sight of the laser inclined 40° off-zenith. The detection of the Doppler shift is performed by means of a double-edge Fabry-Perot interferometer (FPI), as detailed in the following section.

The general design of transmitter-receiver system is shown in Fig. 1. The transmitter of the lidar is based on a Quanta-Ray Pro290 Q-switched, injection-seeded Nd:YAG laser emitting at 532 nm with a repetition rate of 30 Hz and 800 mJ per pulse energy. In seeded mode, the linewidth of the laser beam is less than 0.003 cm$^{-1}$ (90 MHz at 532 nm), whereas the shot-to-shot frequency stability is better than 10 MHz (RMS). During the measurement, the laser beam is commuted successively to each of the three fixed mosaic telescope assemblies, respectively zenith (1), North (2) and East (3), using a galvanometric scanner mirror (4). Each telescope assembly has a field of view of 0.1 mrad and is comprised of a central transmitter shaft with a beam expander (5), ensuring the beam divergence of 35 µrad (full angle at half maximum) and four collecting parabolic mirrors of 500 mm diameter (6), which translates to the total collective area for each telescope of 0.78 m$^2$.

The backscattered light is collected by means of 200 µm multimode optical fibers located at the focal point of each mirror (1500 mm focal distance) and linked to an optical commutation chamber (7), which transfers the collected light through a 600 µm fiber from a given telescope to the entrance of the spectral analysis sub-system. The latter (not shown) comprises the FPI etalon in a thermally-stabilized pressure-controlled chamber, a 0.3 nm interference filter for reducing the sky background and a mode scrambler, which serves for homogenizing the incidence angles of light projected onto the FPI. The homogeneity of the flux angular distribution is important because the transmission function of the FPI depends on the angular incidence. The scrambler module comprises two lenses, the first collecting the light from the input fiber and the second projecting its image onto the output fiber.

The detection of the spectrally-processed light is done with two pairs of cooled super-bialkali Hamamatsu R9880-110 photomultipliers (PMTs), receiving respectively 95% and 5% of the flux (high- and low-gain channels). The high-gain PMTs are electronically gated at 100 µs, i.e. 15 km radial distance. The acquisition is done using a four-channel Licel transient recorder featuring 32760 gates of 50 ns width (i.e. 7.5 m radial resolution).

The upgrade of the lidar (with respect to the design described by Souprayen et al. (1999a)) was carried out during 2012-2018 period and included replacement of various parts. The essential improvements that allowed extending the vertical range for wind profiling are due to the following upgrades: a higher-power laser (24W versus 10W), a new interference filter (0.3 nm vs 1 nm), and the new PMTs with faster response and lower dark current. Additionally, a new Licel transient recorder (50 ns versus 1µs gate bins) and a new cooling system have been introduced.

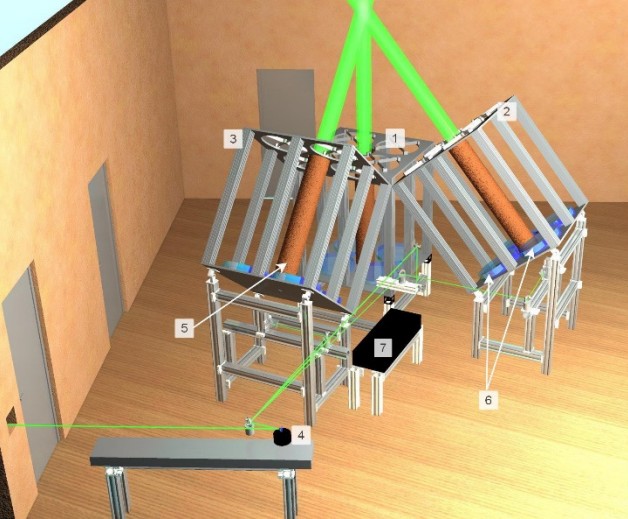

**Figure 1. Three-dimensional representation of the transmitter-receiver system of LIOvent lidar comprising three telescope assemblies for zenith (1), north (2) and east (3) lines-of-sight with four collecting mirrors (6) and a beam expander (5) each. The**
115 **emitted laser beam is commuted between the telescopes using rotating scanner mirror (4). The backscattered light collected by each mirror is transferred via fibers into the optical commutation chamber (7) for aggregating the fluxes from the four mirrors of each telescope assembly.**

### 2.2 Doppler shift detection

The detection of Doppler shift is done using a double-edge Fabry-Perot interferometer (FPI). The FPI etalon (manufactured by StigmaOptique) is assembled by molecular contact and features two half-disc areas with slightly different air gaps, which results in two distinct bandpasses on the both sides of the Rayleigh-Mie backscattered line, as shown in Fig. 2. The Doppler shift of the backscattered line (shown as dashed black in Fig. 2) enhances the signal transmitted through the channel A whilst reducing that of the channel B. The interferometer is set in a sealed pressure-controlled chamber, allowing a
spectral tuning of the FPI A and B bandpasses relative to the backscattered line through variation of the air-gap refractive index. Depending on the atmospheric temperature, the backscattered Rayleigh line obtains a FWHM between about 2 and 2.4 pm, whereas the Mie line is assumed to have the same spectral width as that of the laser (<0.08 pm). The spectral spacing of

the FPI A and B bandpasses of 5.24 pm is determined by the difference in optical thickness of the respective half-disc areas of the interferometer (34.5±0.1 nm), whereas the FWHM of the FPI bandpasses depends on its finesse and amounts to 2.03±0.01 pm according to a series of experiments by Souprayen et al., (1999a,b).

The Doppler shift response profile $R(z, \theta)$ for a given line of sight is calculated as:

$$R(z, \theta) = \frac{C N_A(z,\theta) - N_B(z,\theta)}{C N_A(z,\theta) + N_B(z,\theta)} \ , \tag{1}$$

where $N_A(z, \theta)$ and $N_B(z, \theta)$ are the number of photons received from altitude $z$ and transmitted through the bandpasses A and B respectively; and $C$ is the corrective factor accounting for a possible imbalance between the signals in channels A and B due to a difference in detectors' sensitivity. The corrective factor corresponds to the ratio between the channels A and B and is obtained by comparing $N_A$ and $N_B$ signals from a continuous white source. The Doppler shift (in units of pm) is deduced from the response profile through the instrumental calibration function, which accounts for the temperature broadening of the Rayleigh backscattered line.

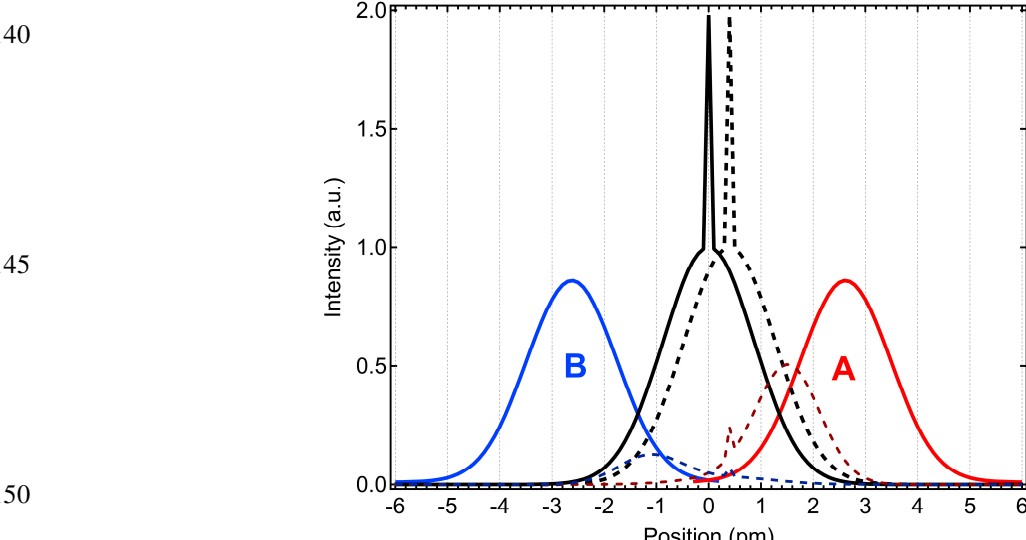

**Figure 2. Spectral shapes of the thermally-broadened Rayleigh backscatter line with the thin Mie line on top of it (solid black) and the FPI bandpasses A and B (solid red and blue). The Doppler-shifted backscatter line (corresponding to an imaginary wind speed of 175 m/s) is shown as black dashed curve. The signals transmitted through the A and B bandpasses for the Doppler-shifted backscattering are illustrated by the dashed red and blue curves.**

The procedure of FPI calibration is thoroughly described by Souprayen et al. (1999b). Briefly, it makes use of the pressure scanning system, which allows for spectral sampling across the two FPI bandpasses by sequentially shifting their spectral position with respect to the spectrally-stable laser line. With the constant and known spectral spacing between FPI bandpasses, one can relate the scanner motor steps to the unit of pm. This relation is then used to retrieve the FWHM of each bandpass,

which together with the FPI spectral spacing and temperature-dependent spectral width of the backscattered line yields the instrumental calibration function. The calibration function is linear in the central zone (between -0.2 and 0.2 pm) and obtains the slope of 0.755 $pm^{-1}$ at 210 K. The uncertainty in the FPI characteristics induces an uncertainty of ±0.3% on the horizontal wind velocity, whereas the effect of temperature uncertainties does not exceed 0.07% per Kelvin (Souprayen et al., 1999a). Since the FPI is placed in a sealed chamber, its spectral characteristics remain constant with time and the calibration through pressure scanning is needed only in case of upgrade the laser unit or the optical processing box.

An absolute measurement of the wind velocity requires a careful determination of the null Doppler shift reference, which is done through 1-minute zenith-pointing acquisition within each 5-minute cycle. This enables accounting for the possible drift in the emitted laser wavelength, typically of 0.03-0.08 $pmh^{-1}$. The horizontal wind components are then obtained by subtracting the line-position profile for the vertical pointing $\overline{P(z, 0°)}$, from those of the tilted pointings $P(z, 40°)$:

$$v_h(z) = \frac{c}{2\lambda_0 \sin 40°}\left[ P(z, 40°) - \overline{P(z, 0°)} \right],\tag{2}$$

where $\lambda_0$ is the emitted wavelength; and $\overline{P(z, 0°)}$ is the reference (null Doppler-shift) position given by an average in the altitude range of 15-25 km, a region where the vertical wind velocity is negligibly small. Expectedly, the line-position profile for the vertical pointing does not vary with altitude therefore the resulting wind profiles are insensitive to the choice of the vertical range for the null Doppler shift reference.

The spectral tuning of the FPI bandpasses with respect to the laser wavelength is verified at the beginning of each measurement session through adjustment of the air pressure inside the FPI chamber using a stepper motor. The temperature inside the FPI housing module is maintained at 30° C at all times and we note that over many months of lidar operation, the spectral tuning remains stable, except after an occasional laser maintenance.

### 2.3 Signal processing and statistical uncertainty

The measurement cadence is such that the zenith, north and east lines of sight are alternated in a cycle of 1-2-2 minutes respectively. A typical acquisition lasts 5 hours during nighttime, that is 2 h integration for each tilted pointing, which ensures signal-to-noise ratio better than 2 all the way up to about 80 km altitude a.s.l. Figure 3a shows an example of raw lidar return profile from the North pointing obtained by stitching the low- and high-gain signals. The vertical range of the useful signal spans between about 5 and 80 km. The lower boundary is due to strong returns from the lower troposphere saturating the detectors in addition to an incomplete geometrical overlap below 2 km.

The photons received from the transient recorder are aggregated over 1-minute intervals and downsampled to 1 μs bins (150 m radial resolution). The off-line signal pre-processing includes subtraction of background due to sky light and PMT thermal noise as well as dead-time correction, after which the response profiles are calculated for each line-of-sight according to Eq. (1). Then, the Doppler shift (line-position profile) is computed using the instrument calibration function with account for atmospheric temperature profile, provided by operational analysis by European Center for Medium-range Weather Forecast

(ECMWF). Finally, the zonal and meridional wind components are obtained by comparing the tilted East and North pointings to the corresponding zenith pointing using Eq. (2).

Statistical error due to Poisson noise (shot error) increases with altitude proportionally to the exponential decay of molecular backscatter. As the error scales with $1/\sqrt{\Delta z}$ (where $\Delta z$ is the vertical resolution), we use a height-dependent $\Delta z$, which is set to 115 m (150 m radial) below 25 km and then increased quasi-exponentially with altitude, from 500 m at 40 km to 4000 m at 70 km (Fig. 3b). For a given vertical resolution, we compute the statistical error (m/s) of an individual response profile:

$$\sigma_R = 2C \frac{\sqrt{N_A(N_B - Fc_B)^2 + N_B(N_A - Fc_A)^2}}{[C(N_A - Fc_A) + (N_B - Fc_B)]^2}, \tag{3}$$

where $Fc_A$ and $Fc_B$ is the background signals in channels $A$ and $B$ and $C$ is the corrective factor introduced in the previous section.

Figure 3c shows the altitude profiles of statistical error for different integration times. For a typical lidar acquisition lasting 5 hours (i.e. 2 hours of a given tilted pointing acquisition, blue curve), the statistical error is less than 2 m/s below 33 km and does not exceed 6 m/s throughout the stratosphere. In the mesosphere, the error increases from 6 m/s at 55 km to 16 m/s at 70 km. A longer acquisition (13.8 h, red curve) reduces the error yet does not extend the vertical coverage: at 75 km altitude, the statistical error for 5 h and 13.8 h acquisition are nearly the same. We use the statistical error value of 25 m/s as a threshold for cut-off altitude of retrieved wind profiles. Given such threshold, the top of vertical range for a standard (5 h) lidar acquisition is ~75 km, whereas a 5-minute acquisition (pink curve), corresponding to a single north-east-zenith measurement cycle, enables coverage up to about 44 km.

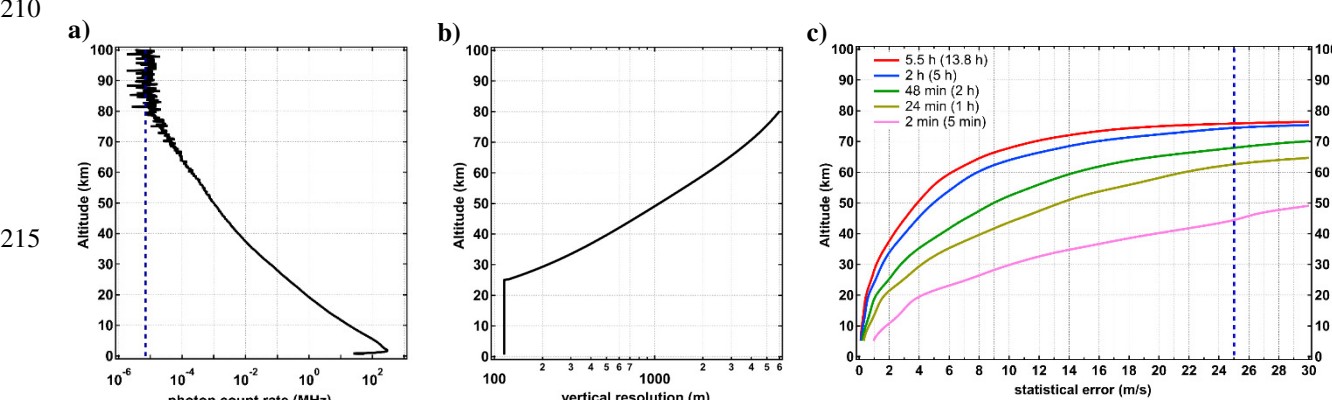

**Figure 3. (a): nightly-average raw signal vertical profile obtained in a 7-hour lidar acquisition on 28.01.2019. Dashed blue curve indicates the background noise level. (b) altitude-variable vertical resolution used in the retrieval. (c) statistical error profiles computed for different acquisition times of a tilted (north) pointing. The values in brackets correspond to the total duration of lidar acquisition, including zenith, north and east pointings in cycle of 1-2-2 minutes. The dashed blue line marks the cut-off error threshold.**

## 3 Comparison with collocated radiosoundings

Over the 4-yr period, spanning June 2015 to June 2019, the validation of the LIOvent wind lidar was conducted using 12 radiosonde (RS) ascents performed at OHP during the time of lidar acquisition. We used Meteomodem M10 radiosondes equipped with GNSS receiver, launched under TOTEX 1200 gr weather balloons. The balloons were reaching on average 29.9 km altitude, whereas the horizontal drift during ascent did not exceed 90 km from the launch point.

Figure 4 displays the altitude-coded trajectories of the 12 radiosonde ascents as well as the ground projections of LIOvent tilted pointings. The horizontal displacement of the radiosondes with respect to the lidar sampling location at every altitude level was calculated separately for the East and North pointings and amounted respectively to 27±19.5 km and 39±25 km (1σ), the largest being 117 km for the North pointing. Generally, the displacement increased with altitude as the balloons were drifting away from the lidar sampling locations, as Fig. 4 suggests.

**Figure 4. OHP wind lidar sampling location along north and east lines-of-sight (thick lines) and trajectories of 12 radiosondes launched from OHP for wind lidar validation (thinner curves) color-coded by the altitude a.s.l. Grey circles indicate the distance from OHP. Particular radiosonde flights are tagged by white arrows with indication of the flight date. The magenta line shows the ground track of Aeolus lidar (see Sect. 5).**

For setting up the intercomparison, lidar measurements were averaged over the time period of radiosonde ascent (110 minutes from the ground to 33 km altitude at 5 m/s ascent rate), whereas the radiosonde measurements, reported at 1 Hz frequency, were downsampled to match the vertical resolution of lidar profiles (115 m – 320 m depending on the altitude). The results of intercomparison are reported in Table 1 as absolute difference between RS and LIOvent wind profiles, standard deviation of the differences and correlation for each particular sounding. The intercomparison exercise is done separately for the zonal and meridional wind components as well as for the total wind speed and wind direction.

The mean differences obtained from the individual comparison cases varies between -1.3 and 0.9 m/s for the zonal wind, and between -2 and 0.9 m/s for the meridional wind. For the total wind and direction, the differences vary between -1.1 and 0.7 m/s and between -4.9 and 9.6 degrees respectively. The averages of all intercomparison cases amount to +0.1 m/s and -0.1 m/s respectively for the zonal and meridional components, 0.0 m/s for the total wind and 0.3 degrees for the wind direction.

Figure 5 (top panels) shows altitude profiles of the absolute difference between LIOvent and RS for each sounding as well as the median profile. The point-by-point differences rarely exceed 5 m/s, whereas the median values never exceed 2 m/s. While the median difference profiles do not indicate any altitude-dependent bias, the scatter of differences appears to increase with altitude. This is due, on one hand, to larger horizontal offset between measurements at higher altitudes and, on the other hand, due to increase of statistical error with altitude shown as dashed curves in the a) and b) panels of Fig. 5.

The bottom panels of Fig. 5 display the scatter plots of the wind velocities measured by the lidar and the radiosondes with associated regressions and correlation coefficients. For both wind components, the slope of the regression line is close to 1, which affirms the credibility of the FPI calibration function relating the Doppler shift response to wind velocity. The Pearson correlation coefficient $r$ deduced from the ensemble of collocated measurements amounted to 0.97 and 0.96 respectively for zonal and meridional wind velocities, 0.97 for the total wind and 0.89 for the wind direction.

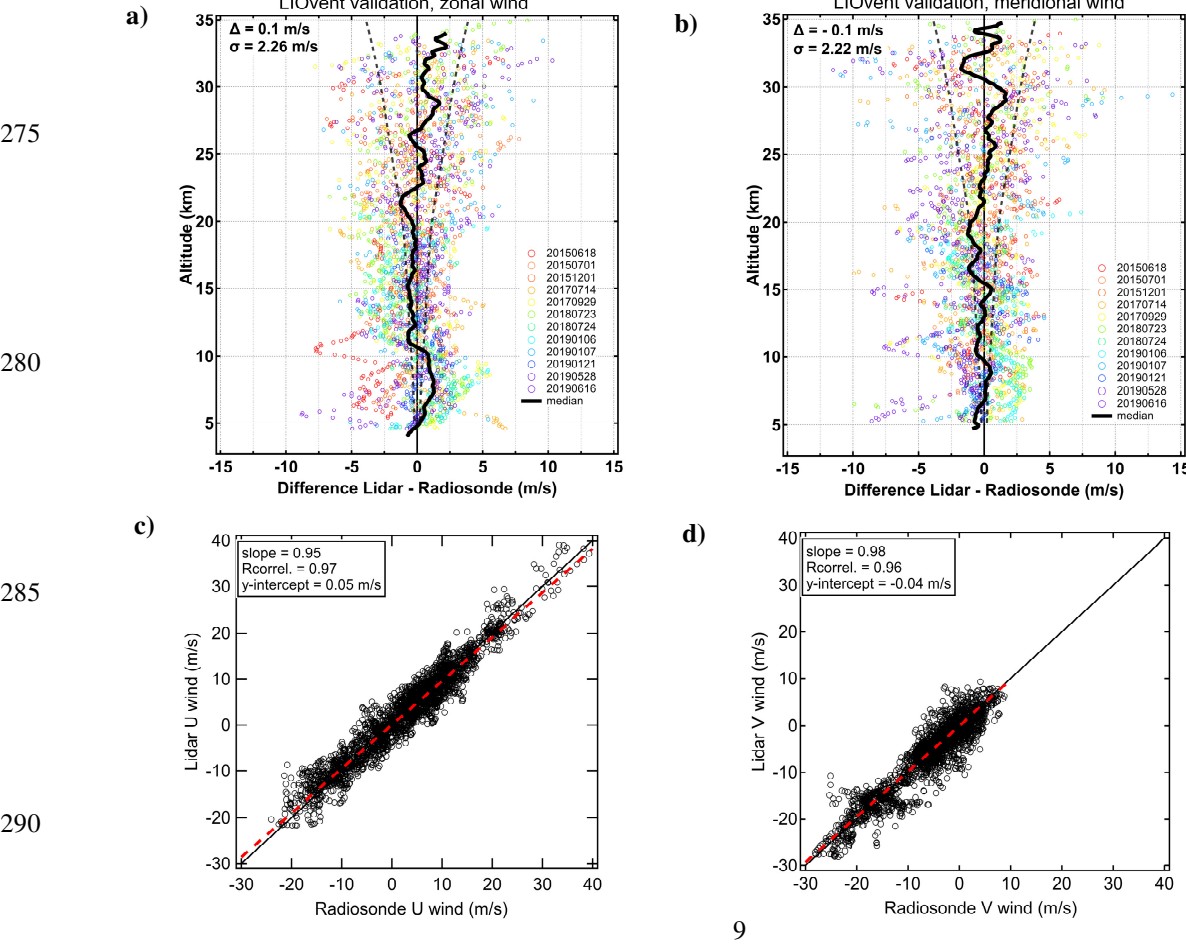

**Figure 5. Summary of LIOvent lidar validation using radiosonde ascents from OHP. Top panels: absolute difference between LIOvent and radiosonde zonal (a) and meridional (b) wind velocity for each sounding (date-colored circles, dates provided in the legend) and median profile (black curve). The dashed line indicate the statistical uncertainty estmated for a 2-hour lidar acquisition. The mean difference Δ and the mean standard deviation of the difference (σ) are indicated in the top-left corner of a) and b) panels. Bottom panels: scatter plots of the zonal (c) and meridional (d) wind velocities measured by the lidar and the radiosondes. The 1:1 line is shown in solid black, the linear regression line is shown in dashed red.**

The mean standard deviation of the differences for the 12 collocated soundings amounts to 2.26 m/s for the zonal and 2.22 m/s for the meridional wind profiles. These values are consistent with the estimated shot error for a 2 hours lidar acquisition (i.e. duration of a radiosounding), which increases from 0.2 m/s to 3.4 m/s in the altitude range of lidar-radiosonde intercomparison, as can be inferred from Fig. 5a,b. For evaluating the effect of the horizontal offset between the lidar and RS measurements we computed the offset-weighted averages of the intercomparison statistics and compared them with the ordinary averages. The weight for each individual value is defined as $w = 1 - \overline{D}/D_{max}$, where $\overline{D}$ is the mean distance between the lidar and RS sampling locations and $D_{max}$ is the maximum distance amounting to 69 km (Table 1). We note that the horizontal-offset weighting of the differences neither affects the mean difference nor the mean correlation but reduces the standard deviation for the wind components and total wind by about 0.2 m/s. This suggests that horizontal variability of the wind field on a scale of few tens of kilometres is small yet non-negligible.

Figure 6 presents examples of summer and winter zonal wind intercomparison cases, both showing directional wind shear in the lower stratosphere but of opposite sign. Remarkably, the small-scale fluctuations, presumably caused by gravity waves propagation and/or breaking, are reproduced by the lidar just as accurately as measured by the balloon sonde, carried by those winds.

At higher altitudes, the fine-scale fluctuations resolved by the lidar appear at times out of phase with those seen by the radiosonde. This is more prominent in the Summer case (Fig. 6a) despite the closer collocation of the measurements. In this case, the RS ascent closely follows the lidar's line-of-sight up until 19 km (cf. Fig. 4) while the LIOvent zonal profile precisely tracks the one of RS up to about the same level. At 19 km, the zonal wind reverses, the balloon makes a U-turn and progressively drifts westward and away from the lidar. Above 30 km, the RS and LIOvent profiles start to get out of phase whilst both showing an increasing easterly wind between 30 – 35 km. The lidar profile, extending above the top of radiosounding, reveals a typical signature of a gravity wave, supposedly propagating in the zonal direction (considering a relatively unperturbed meridional wind profile in this layer, not shown).

While the statistical error of the lidar measurement becomes comparable to the observed variations at these levels, the dephasing of LIOvent and RS profiles in Fig. 6a is likely due to spatially-offset sampling of the gravity wave front. Interestingly, the dephasing above 30 km is less obvious in the winter case (Fig. 6b) despite a considerably larger spatial offset, compared to the summer case (88 km vs 36 km). This may be explained in consideration of the much stronger zonal wind in the winter case (38 m/s versus -2 m/s), damping the amplitude of the wave-induced perturbations.

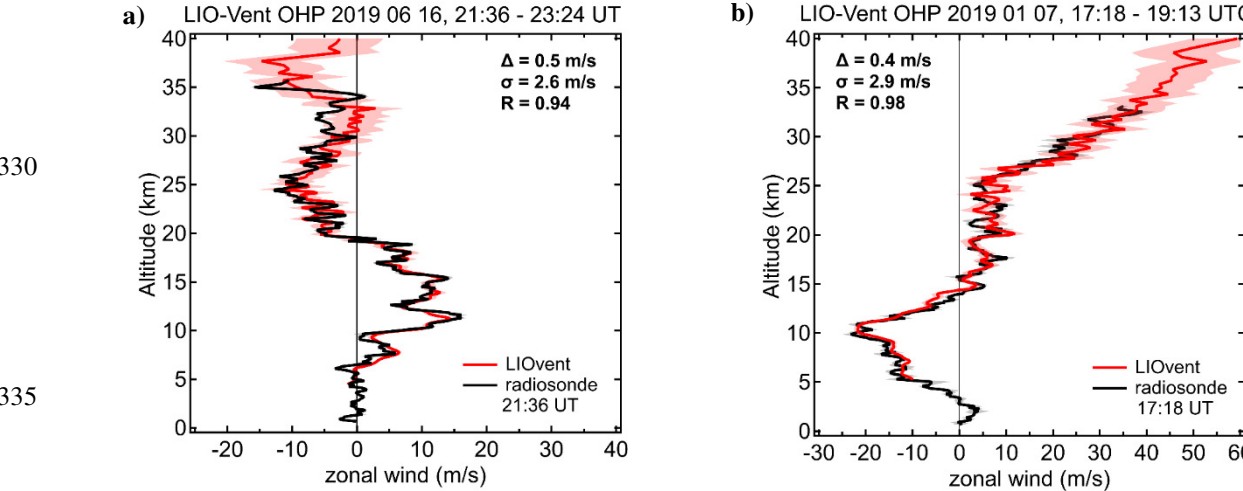

**Figure 6. Selected cases of the lidar-radiosonde intercomparison of the zonal wind profiles in June (a) and January (b) 2019. The lidar measurement dates and times are given above the panels, the time of radiosonde launch on the same date is provided in the legend.**

## 3.1 Sensitivity to Mie scattering

Although the Mie-backscattered line is narrow (0.08 pm) compared to the thermally-broadened Rayleigh line (2 – 2.4 pm) the intensity of the former may be substantially higher and thereby alter the spectral shape of the return signal. In this case, a disproportionally larger flux would be transmitted through one of the FPI bandpasses, affecting its calibration function and introducing a bias into the wind retrieval within the particle layer. The sensitivity to Mie scattering can be reduced by increasing the FPI spectral spacing, however this also reduces the sensitivity to the Doppler shift. The optimal spectral configuration of the FPI has been established on the base of a theoretical model by Souprayen et al. (1999b). They found that for observable stratospheric wind velocities, the residual Mie-induced error is less than 1 m/s for the scattering ratio $R$=10, which is characteristic of a cirrus cloud readily visible to an unaided eye.

In this study, we have experimentally revisited the aspect of FPI sensitivity to particulate scattering. The eruption of the Raikoke stratovolcano (22 June 2019, Kuril Islands, 48° N, 153° E) has polluted the lower stratosphere with a large amount of sulfuric aerosol (NASA EarthObservatory, 2019). The aerosol plumes were observed by OHP lidars every night since 10 July 2019 (and at the time of writing) between 12 and 20 km altitude, which provided an opportunity for testing the sensitivity of wind lidar to Mie-scattering in the stratosphere.

Figure 7 displays lidar measurements of aerosol scattering ratio (SR) and zonal wind velocity on 20 July. The SR profiles were obtained from an aerosol channel (532 nm) of LTA (Lidar Temperature Aerosol) instrument (Keckhut et al., 1993) and from the zenith acquisition of LIOvent lidar using aerosol retrieval method described by Khaykin et al. (2017 and

references therein). The LIOvent operation was started after the end of LTA operation since the lidars share the same laser and cannot be operated simultaneously. Both lidars consistently show an aerosol layer at 16.2 km altitude with SR reaching 4.7 and an estimated optical depth of 0.03 which is comparable to a thin cirrus cloud (Hoareau et al., 2013). In addition, the LIOvent measurement reveals a cirrus cloud at 12.2 km, which occurred only towards the end of LTA acquisition and thereby left a weaker imprint in the average SR profile of LTA.

The LIOvent wind measurement in the presence of ice crystals and volcanic aerosol is compared in Fig. 7 to a time-collocated radiosoundings conducted from Nimes airport, situated ~100 km west from OHP (cf. Fig. 4). While the vertical structures in the LIOvent and RS wind profiles are at times out of phase (which may be explained by spatial variability), the lidar profile does not show any indications of Mie-induced bias, neither due to a thin cirrus cloud nor due to a volcanic aerosol layer. Such a bias would appear as a sharp feature in the wind profile coinciding the with the SR enhancement, which is obviously not the case here. This result confirms that the spectral configuration of the FPI allows accurate wind measurements in the presence of particles in the middle atmosphere.

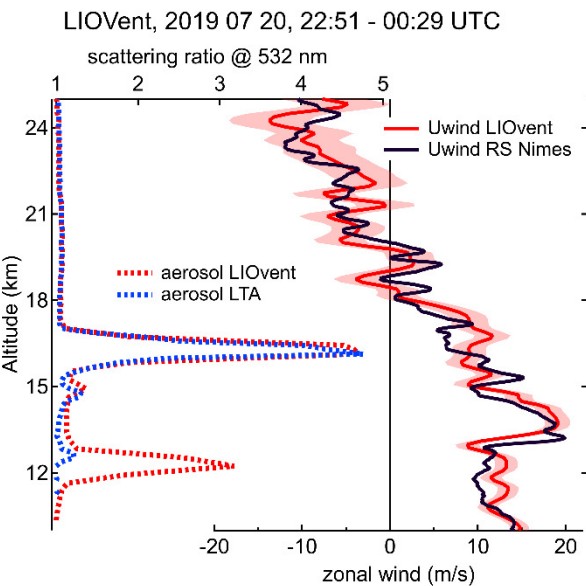

**Figure 7. Zonal wind profiles (bottom axis) measured by the LIOvent wind lidar (red solid) and by MeteoFrance routine radiosounding launched from Nimes airport (100 km away from OHP, see Fig. 4) at 00 UT on 21.07.2019 (black solid). The date and time of LIOvent measurement is shown at the panel top. The aerosol scattering ratio profiles obtained using LIOvent zenith pointing (red dashed) and LTA lidar (blue dashed), showing volcanic aerosol layer at 16.2 km are plotted versus top axis. See text for details.**

## 4 Observations

During the 2015-2019 period, the LIOvent instrument was operated on 52 nights, mostly during early summer and winter seasons. This section reports examples of successive nightly-mean profiles reflecting the wind variability in the middle atmosphere during opposite seasons as well as a particular case of temporally-resolved wind profiling.

### 4.1 January 2019 series

        An interesting dynamical event in the USLM was observed in January 2019 during an intensive measurement campaign

dedicated to Aeolus validation (AboVE-OHP, Aeolus Validation Experiment at OHP). A strong perturbation of the Arctic circumpolar vortex has occurred as a result of a major Sudden Stratospheric Warming event during the first week of January 2019. According to potential vorticity maps (not shown) based on ECMWF Integrated Forecast System (IFS), the vortex started to split around 1[nd] January and evolved into two separate vortices above Europe and Canada by 4[th] January. The European counterpart was displaced southward and its edge - at 850 K potential temperature level (~50 km) - reached OHP around 6[th]

January, that is when the AboVE-OHP measurement campaign was started.

        Figure 8 shows ensembles of the zonal and meridional wind profiles obtained during 6 - 9 January period. The plots include indications of the stratopause level which was progressively descending from 47 to 43 km during that period, as inferred from simultaneous temperature profiling using LiO3S (Lidar O3 stratosphérique) differential absorption lidar (Godin-Beekmann et al., 2003; Wing et al., 2018). The 6[th] January wind profiles (red curves) reflect the perturbed conditions when

the vortex edge was located above OHP and both zonal and meridional components were maximizing at 80 m/s around the stratopause. As the edge of vortex was moving back north of OHP during the following days, the measurements show weakening winds throughout the USLM and reversal of both wind components in the lower mesosphere by 9[th] January. The rapidly weakening winds form an envelope of profiles with a bottom at ~27 km for the zonal wind and ~38 km for the meridional component. Below this envelope, neither of the wind components show significant change over the 4-day period.

The observed evolution of wind profiles is reproduced by the ECMWF T1279/L137 operational analysis represented by cross-circles in Fig 8. The wind change envelope and the vertical structure of the wind profiles are both well resolved by the model. The ECMWF profiles reproduce the observed vertical fluctuations on a scale of a few kilometres up to the stratopause, which is remarkable since the regular radiosoundings assimilated into the model hardly reach 30 km altitude. We note that the consistency between ECMWF and LIOvent is better for the zonal wind, whereas the vertical structures in

meridional wind are somewhat less consistent with the observations in the USLM. Analogous results, inferred from intercomparison between ALOMAR wind lidar and ECMWF forecast winds, were reported by Rüfenacht et al. (2018). The damping and dephasing of the vertical structures by ECMWF becomes more prominent above about 50 km, which might owe to both the temporal averaging over 5-13 hours by the lidar and the coarse model resolution in the mesosphere. A detailed comparison between wind lidar observations, ECMWF IFS and reanalysis data will be the subject of a separate study.

**4.2 May 2019 series**

Figure 9 shows a sequence of wind profiles acquired during late May 2019. While the winds appear highly variable in the upper troposphere due to the dynamics of the jet stream, the middle stratosphere remains relatively calm and stable over the considered 4-day period. The zonal wind reverses at around 36 km and the easterlies pick up until ~65 km, that is when the wind shear reverses. The meridional wind is very weak throughout the stratosphere except for a small envelope at 30 km, which accompanies the zonal wind perturbations in this layer. The ECMWF IFS accurately resolves this envelope, however the observed vertical structures above, in the USLM, are not reproduced by the model. In particular, the reversal of wind shear at around 65 km in both wind components is missed by the model, whereas the lidar profiles consistently report this feature, significant at the permitted statistical error of 25 m/s.

The imposed error threshold of 25 m/s determines the cut-off altitude of the wind profiles reported in Figures 8 and 9. The top altitude varies between 65 and 75 km depending on the presence of cirrus clouds inhibiting the return signal from above. We note that the meridional profiles normally reach higher altitudes, which is due to a better condition of the collecting mirrors of the north-pointing telescope assembly.

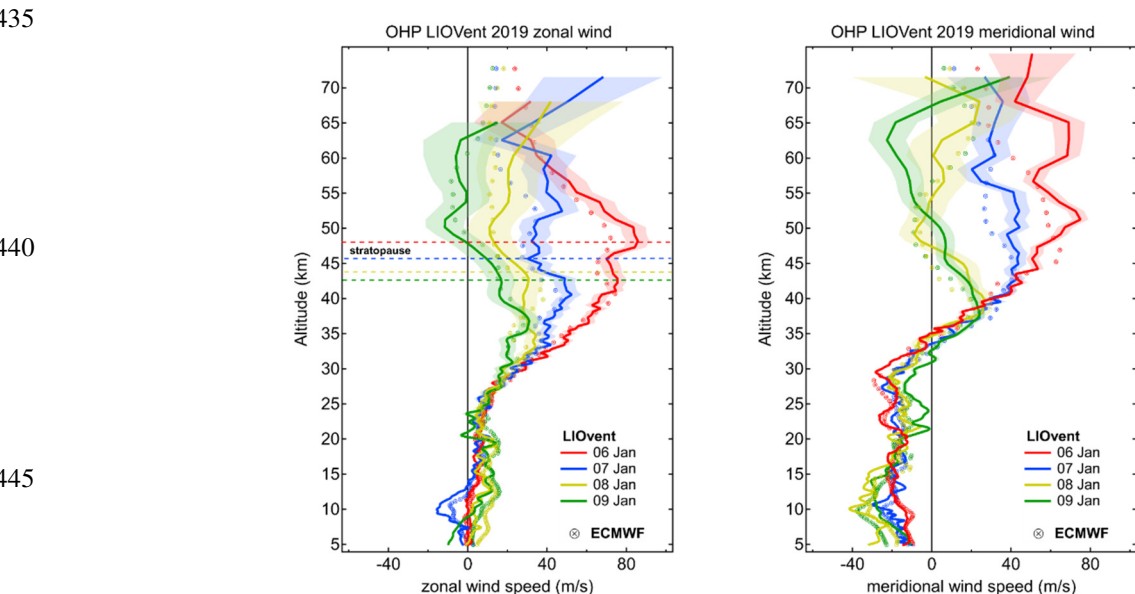

**Figure 8. Ensembles of nightly-mean vertical profiles of zonal (left) and meridional (right) wind profiles obtained by LIOvent in January 2019 (solid curves) with statistical uncertainty shown as shading and the corresponding ECMWF Integrated Forecast System profiles (cross-circles). Horizontal dashed lines in the left panel indicate the stratopause altitude obtained from simultaneous temperature lidar measurements.**

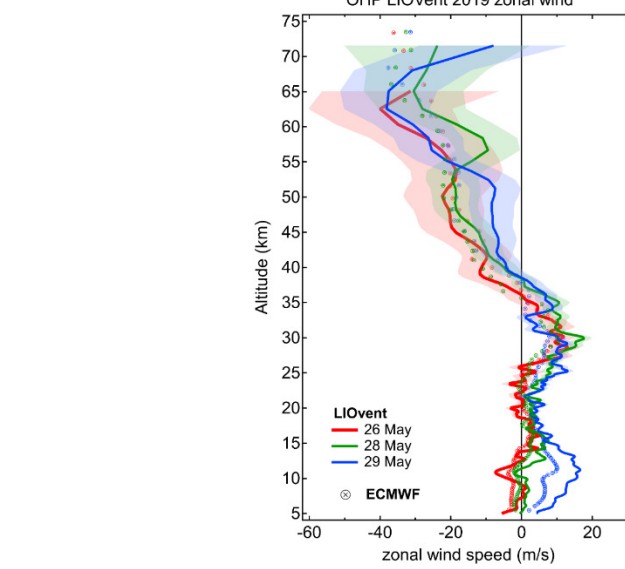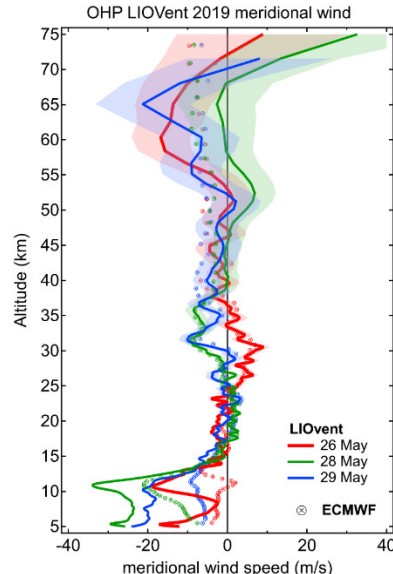

**Figure 9. Same as Fig. 8 but for May 2019**

### 4.3 Time-resolved wind profiling

An important advantage of the Doppler lidar technique is the capacity to provide temporally-resolved vertical profiling of the atmosphere, which enables characterization of high-frequency fluctuations in the wind profile, inaccessible with snapshot-like radiosonde measurements. Figure 10a provides an example of meridional wind profile variation over the course of a continuous whole-night lidar acquisition lasting nearly 14 hours. Superimposed onto the lidar time-altitude section is a radiosonde ascent, plotted using the same color map as the lidar wind curtain.

The LIOvent and RS profiles are remarkably consistent as can be seen from the color map (correlation coefficient amounts to 0.99 in this case). With that, the lidar wind curtain shows important variation of the wind velocity over the course of 14-hour acquisition. The peak-to-peak variation at any level below 30 km altitude is between 10 and 20 m/s, increasing to ~30 m/s towards 40 km. The wind change rate in any 3 km thick layer is reaching 10 m/s per hour, which points to the predominance of temporal variability of the wind field over its spatial variability. Indeed, the maximum deviation of the lidar profile from the RS one in this case did not exceed 4 m/s at any given level, all the while that the RS measurements were taken as far as 71 km away from the lidar sampling location (cf. Fig. 4).

In the upper-middle stratosphere (i.e. around 35 km), where the meridional wind reverses, one can discern wind patterns slowly propagating downward. This pattern is a typical signature of upward-propagating gravity waves with a non-zero ground-based phase speed. A somewhat different pattern is observed in the lower-middle stratosphere (15 – 30 km), where the vertical structures appear to remain constant with altitude. Figure 10b provides a deeper insight into the variable structure of

this layer by showing a sequence of 6 wind profiles, obtained by integrating over successive 135-minute temporal intervals as well as the corresponding RS profile. One can see three layers of stronger southward wind at around 17 km, 23 km and 30 km altitude interleaved by two layers of weaker wind at around 20 and 26 km.

The persistence of the observed structures in both temporal and vertical dimensions suggests the occurrence of stationary gravity waves, most likely generated by the flow over the Alpine mountains. Indeed, the circulation in the lower troposphere at that time (not shown) was such that the OHP site was downwind of the Alps. The stationary gravity waves, generated by the flow over the mountain range, could propagate freely into the stratosphere because of the absence of directional wind shear all the way up to 35 km. The amplitude of these wave-induced perturbations appears to increase with altitude from ~5 m/s to ~10 m/s, which is expected from the linear theory of atmospheric waves.

The orographic nature of the gravity wave, identified using time-resolved lidar measurements, can be verified in consideration of the vertical wavelength. For a stationary wave, the vertical wavelength $\lambda_z$ can be deduced from the horizontal wind speed $v_h$ and the buoyancy frequency $N$:

$$\lambda_z = 2\pi \frac{v_h}{N} \tag{4}$$

Given the observed $v_h$ ~20 m/s and $N$~0.02 s$^{-1}$, we obtain the vertical wavelength of ~6.5 km, which corresponds well to what can be deduced directly from the wind profiles in Fig. 10.

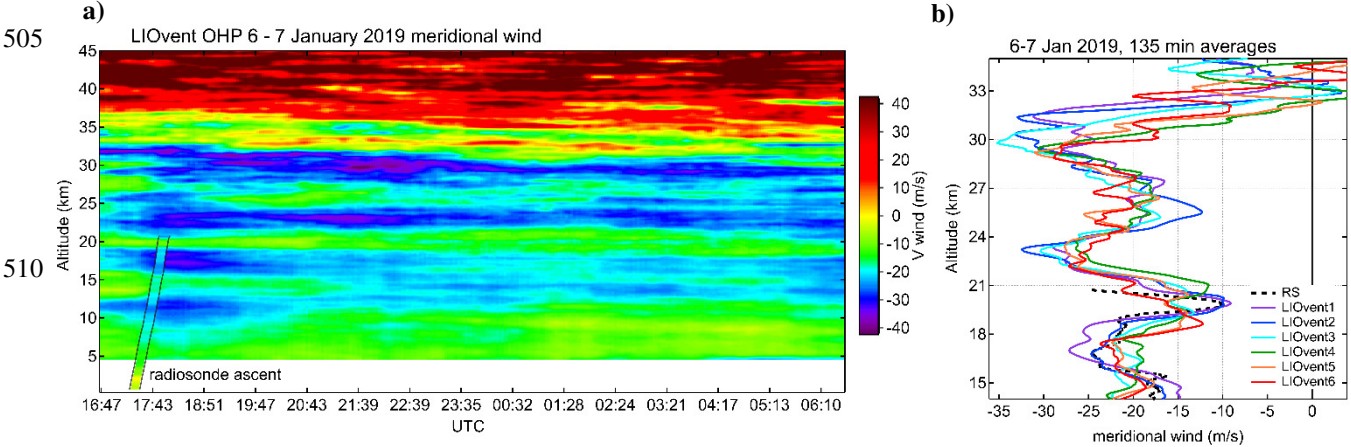

**Figure 10. (a) Temporal variation of the meridional wind profile over the course of a continuous whole-night LIOvent acquisition started on 6 January 2019. Superimposed onto the lidar time-altitude section is the corresponding radiosonde ascent from OHP, plotted using the same color map as the lidar wind curtain. (b) successive 135-minute averages of meridional wind measured by the lidar (solid curves) and the radiosonde profile (black dashed).**

**5 First results of Aeolus validation**

Aeolus is the ESA's satellite mission designed to measure wind and aerosol profiles in the troposphere and lower stratosphere on a global scale (Stoffelen et al., 2005; ESA, 2008). Launched on 22 August 2018, the Aeolus satellite carries the Atmospheric LAser Doppler INstrument (ALADIN) which features a telescope of 1.5 m diameter and a laser emitting at 355 nm with a repetition rate of 50 Hz and ~65 mJ per pulse energy (Reitebuch et al., 2019). ALADIN instrument has two detection channels for measuring Doppler shift using the molecular (Rayleigh) and particulate (Mie) backscattering. The

Rayleigh channel makes use of a double-edge Fabry-Perot interferometer, that is the measurement principle exploited by the OHP wind lidar.

The ALADIN telescope is pointed 35° away from the orbital plane in order to sense the backscattered light perpendicular to the trajectory of the satellite. This enables measuring the so-called horizontal line-of-sight (HLOS) wind velocity, which is close to the zonal wind component except at high latitudes. The Aeolus satellite has a Sun-synchronous

dusk/dawn orbit with a 7-day repeat cycle, passing near the OHP station (within 100 km) twice per week along the successive ascending and descending orbits, at around 17:50 and 5:50 UT respectively.

As Aeolus and the OHP wind lidar exploit the same measurement technique, the LIOvent instrument is an important contributor to Aeolus Cal/Val activity. Since January 2019 and by the time of writing, LIOvent has been operated on 27 nights, providing 8 measurements collocated with Aeolus overpasses. Some of the Aeolus-collocated LIOvent acquisitions were

accompanied by simultaneous radiosonde ascents. While a comprehensive validation exercise will be the subject of a separate study, here we provide an example of comparison between collocated Aeolus level 2B B02 'Rayleigh-clear' and LIOvent wind profiles. One should bear in mind that Aeolus wind data processing is still being improved by optimizing the in-obit instrument calibration, therefore the presented validation case is to be considered as preliminary.

Figure 11 displays a collocation case from 7 January 2019 when the satellite was sampling the atmosphere around 100

540    km west of OHP along the ascending orbit. The plot includes two successive Aeolus wind profiles (blue dashed) obtained by 12 seconds integration (i.e. 90 km along-track distance) as well as their mean (blue solid). The LIOvent and RS wind components are converted to Aeolus HLOS wind and reported at their native vertical resolution (solid red and black) as well as after downsampling to Aeolus vertical resolution (circles).

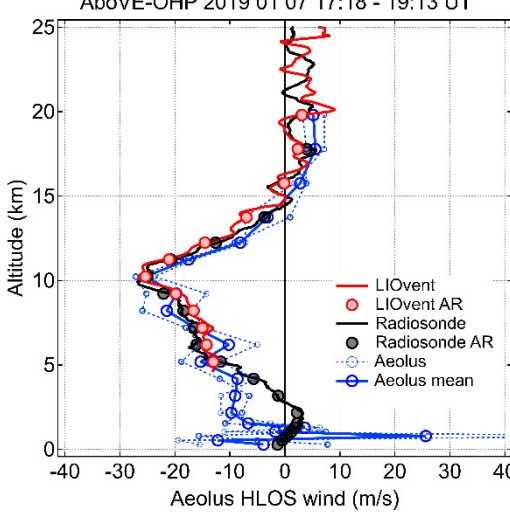

**Figure 11. Example of the validation of preliminary L2B HLOS wind of the ESA Aeolus ALADIN lidar using the LIOvent lidar and a radiosounding within AboVE-OHP campaign. The LIOvent and RS wind components are converted to Aeolus HLOS wind and reported at their native vertical resolution (solid red and black) as well as after downsampling to Aeolus vertical resolution (circles, marked AR in the legend). The Aeolus overpass near OHP took place on 7 January 2019 at 17:50 UT (ascending orbit). The mean distance between LIOvent and ALADIN sampling locations is 106 km. The lidar acquisition time (corresponding to the radiosonde ascent time) is provided in the panel top. See text for further details.**

The profiles are found to be in good agreement, consistently reproducing the peak in the eastward wind of -25 m/s at 10 km, which corresponds to an anticyclonic feature of the jet stream (not shown). In the middle troposphere, successive Aeolus profiles appear somewhat scattered around their mean, with the latter being in better agreement with the ground-based measurements. Below 5 km, the Aeolus profile deviates from the RS, which may be caused by a stronger spatial variability of the wind field in the lower troposphere (note that the minimum horizontal distance between the RS and Aeolus measurements is 91 km). In the lower stratosphere, that is above about 11 km, Aeolus follows well the downsampled measurements by OHP lidar and radiosonde. The average difference between the mean Aeolus and downsampled LIOvent HLOS wind profiles in the overlapping range of 5 - 20 km amounts in this case to +1.5 m/s with a standard deviation of 3.2 m/s and a correlation coefficient of 0.96. Similar values were obtained from other collocations during AboVE-OHP campaign in January 2019.

## 6 Summary and outlook

The OHP wind lidar presented here was a unique instrument at the time of its creation and remains one of the very few instruments capable of wind profiling in the middle atmosphere with vertical resolution up to 115 m and temporal resolution up to 5 minutes. In this paper, we have described the design of the instrument after its upgrade and evaluated its capacities using a dozen time-collocated radiosoundings launched from OHP. We have shown that the lidar is capable of measuring horizontal wind velocity between 5 and 75 km altitude with a random error of less than 6 m/s up to the top of the stratosphere. We note that the vertical range can potentially be extended to 3 – 80 km through replacement of the beam-commuting and beam-splitting mirrors, for which the resources are available.

A noticeable result of the lidar-radiosonde intercomparison is a remarkably small average bias of ±0.1 m/s for the both wind components and 0.3 degrees for the wind direction. This finding affirms the reliability of the on-the-run calibration (through periodical zenith pointing) as well as the stability of the FPI calibration function. Also remarkable is that the small-scale wind fluctuations are reproduced by the lidar just as accurately as measured by the balloon sondes, carried by those winds. The average standard deviation of the differences for the total horizontal wind was found to be only ~2.3 m/s, which is consistent with the error estimates for the considered altitude range. The correlation coefficient obtained from the ensemble of collocated measurements amounted to 0.97 for the total wind speed and to 0.89 for the wind direction.

We have shown that wind profiling with the LIOvent lidar has little or no sensitivity to the presence of particles (thin cirrus clouds or stratospheric aerosols) and we have demonstrated the capacity of the wind lidar to measure vertical profiles of aerosol backscattering. In addition, using the 3 different lines-of-sight, one can obtain information on the fine-scale horizontal variability of stratospheric aerosol.

The examples of successive nightly-mean wind profiles given in Sect. 4.1 and 4.2 provide interesting example of the wind variability in the upper stratosphere and lower mesosphere, the atmospheric layer of exceptionally poor observational coverage. The observed vertical structures and the rapidly changing wind shear reflect the complex dynamics of the USLM layer and its two-way interactions with the upward-propagating gravity waves, whose manifestation may be not well reproduced by atmospheric models. We note though that ECMWF operational model tends to reproduce, in most cases, the observed vertical structures at least up to 50 km altitude.

The example of time-resolved wind profiling presented in Sect. 4.3 highlights the capacity of LIOvent instrument to detect high-frequency fluctuations in the wind profile, indicative of various types of gravity waves. This rare capacity enables a comprehensive characterization of the high-frequency part of the wave spectrum, inaccessible with any other measurement technique. At OHP, the wind lidar acquisitions are typically accompanied by temperature profiling using Rayleigh lidar, which altogether provides a complete suite of thermodynamical parameters in the middle atmosphere on a regular basis and on a long term.

Using the time-resolved wind profiling and simultaneous radiosoundings, we have found that the temporal variability of wind profile in the free atmosphere at a scale of 1 hour may be at least twice as large as the spatial variability on a scale of 50-100 km, as deduced from the lidar-radiosonde intercomparison. This finding is to be considered for Aeolus wind validation activities in a sense that a precise temporal collocation may be more important than the spatial collocation of the wind measurements.

We have presented the first preliminary case of Aeolus validation using the LIOvent lidar. We note that while the Aeolus data processing and calibration may be subject to further improvement, the first results of intercomparison between the ground-based and space-borne Doppler lidars are encouraging. The validation of Aeolus is to be continued at OHP on a regular basis for monitoring of the long-term stability of the satellite lidar, whereas a dedicated Aeolus validation study will be provided in a separate article.

Further studies exploiting LIOvent observations will address the characteristics of gravity waves retrieved from simultaneous wind and temperature profiling at OHP as well as intercomparison with operational analysis and new reanalysis data sets such as ECMWF ERA5. The lidar wind profiling is also to be used in conjunction with infrasound measurements carried out at OHP (Le Pichon et al., 2015) for studies of middle atmosphere dynamics.

***Data availability.*** The wind lidar data will are available through the AERIS data portal at https://cds-espri.ipsl.upmc.fr/NDACC/station?methodName=viewDataOhp. The Aeolus data will be made publically available upon completion of the Cal/Val phase.

*Authors contributions.* SK and AH conceived the study and conducted the LIOvent measurements, RW conducted the LIOvent measurements and extracted Aeolus data, PK offered scientific insight, SGB offered scientific insight and provided access to LiO3S lidar data, JP conceived the optical design of LIOvent instrument, JFM and JS upgraded and optimized the wind lidar. The paper is written by SK with contributions from all co-authors.

*Acknowledgements.* The upgrade of OHP wind lidar was financially supported by CNES (Centre Nationale des Etudes Spatiales) as well as through EU FP7 ARISE and H2020 ARISE2 projects. We thank the personnel of Station Gerard Megie at OHP, Frederic Gomez, Pierre Da Conceicao, Yohann Pignon and others for conducting the radiosonde launches and lidar operation. We thank Alexis Le Pichon (CEA) for providing ECMWF wind profiles for OHP location and Andreas Dörnbrack (DLR) for providing high-resolution PV polar projections based on ECMWF IFS data. The work related to Aeolus validation has been performed in the frame of Aeolus Scientific Calibration & Validation Team (ACVT) activities. Results are based on preliminary (not fully calibrated/validated) Aeolus data, that have not yet been publically released. Further data quality improvements, including in particular a significant product bias reduction, will be achieved before the public data release. Aeolus is a European Space Agency (ESA) mission from the Earth Explorer Program of research satellites. We thank Anne Grete Straume (Aeolus mission scientist) and Jonas von Bismark (Aeolus data quality manager) for the fruitful exchange on this study. We also thank the two anonymous referees for a careful revision of the article.

*Competing interests.* The authors claim no competing interests.

| Date | Mean difference, m/s or degrees (φ) | | | | Stand. deviation m/s or degrees (φ) | | | | Correlation coefficient *r* | | | | Distance, km | | Top of RS, km |
|---|---|---|---|---|---|---|---|---|---|---|---|---|---|---|---|
| | U | V | V$_h$ | φ | U | V | V$_h$ | φ | U | V | V$_h$ | Φ | U | V | Z$_{top}$ |
| 2015 06 18 | -1.3 | 0.1 | -0.9 | 2.9 | 2.9 | 2.3 | 2.6 | 16.8 | 0.95 | 0.96 | 0.97 | 0.93 | 31 | 47 | 32.5 |
| 2015 07 01 | 0.9 | -0.8 | -0.1 | -4.1 | 2.1 | 1.2 | 2.4 | 13.5 | 0.92 | 0.89 | 0.96 | 0.95 | 16 | 21 | 28.0 |
| 2015 12 01 | -0.4 | 0.6 | -0.2 | -1.4 | 1.8 | 2.0 | 1.9 | 18.9 | 0.96 | 0.96 | 0.98 | 0.92 | 15 | 27 | 33.0 |
| 2017 07 14 | 0.1 | -0.3 | 0.5 | 1.9 | 2.6 | 3.1 | 2.5 | 20.3 | 0.98 | 0.79 | 0.99 | 0.96 | 46 | 69 | 33.0 |
| 2017 09 29 | -0.0 | 0.65 | 0.3 | -4.6 | 2.7 | 2.7 | 2.5 | 27.8 | 0.93 | 0.85 | 0.96 | 0.75 | 18 | 38 | 33.0 |
| 2018 07 23 | 0.29 | 0.1 | -0.3 | 0.9 | 3.2 | 2.9 | 2.5 | 19.4 | 0.97 | 0.88 | 0.98 | 0.94 | 21 | 38 | 37.3 |
| 2018 07 24 | 0.2 | 0.0 | 0.5 | -2.0 | 1.6 | 1.3 | 1.5 | 14.6 | 0.99 | 0.89 | 0.99 | 0.97 | 8 | 22 | 20.1 |
| 2019 01 06 | -0.1 | 0.9 | -1.1 | 0.5 | 2.5 | 2.1 | 3.4 | 10.7 | 0.90 | 0.99 | 0.98 | 0.91 | 37 | 45 | 20.9 |
| 2019 01 07 | 0.4 | -0.7 | 0.7 | -0.0 | 2.9 | 3.8 | 3.0 | 8.4 | 0.98 | 0.98 | 0.99 | 0.98 | 57 | 67 | 33.5 |
| 2019 01 21 | 0.1 | 0.0 | 0.2 | -1.7 | 0.9 | 1.5 | 1.2 | 6.7 | 0.99 | 0.98 | 0.99 | 0.99 | 13 | 20 | 18.0 |
| 2019 05 28 | -0.2 | -2.0 | 0.4 | 9.6 | 2.7 | 3.6 | 2.6 | 24.8 | 0.94 | 0.89 | 0.97 | 0.71 | 27 | 49 | 32.5 |
| 2019 06 16 | 0.6 | -0.5 | 0.3 | 1.3 | 2.5 | 2.5 | 2.0 | 20.9 | 0.94 | 0.72 | 0.96 | 0.73 | 12 | 15 | 37 |
| **Average** | **0.1** | **-0.1** | **0.0** | **0.3** | **2.3** | **2.2** | **2.3** | **16.9** | **0.97** | **0.96** | **0.97** | **0.89** | **27** | **35** | **29.9** |

Table 1. Summary of intercomparison between LIOvent lidar and time-collocated radiosoundings launched at OHP. The results are shown separately for zonal (U) and meridional (V) wind measurements as well as for the total wind speed (V$_h$) and wind direction (φ). Provided for each case of intercomparison (from left to right) are: measurement date, mean absolute difference, standard deviation of the differences, Pearson's correlation coefficient *r*, mean horizontal distance between the lidar and radiosonde sampling locations and top altitude of radiosounding. The average values are provided in the bottom row.

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
