# Peer review of "Doppler lidar at Observatoire de Haute Provence for wind profiling up to 75 km altitude: performance evaluation and observations"

_Atmospheric Measurement Techniques, 2019_

## Referee Comment (RC1) · Anonymous Referee #1 · 25 Nov 2019

**Review of Khaykini et al. (2019), AMTD**

**Doppler lidar at Observatoire de Haute Provence for wind profiling up to 75 km altitude: performance evaluation and observations**

**Summary**

After the pioneering work by the OHP group (Chanin et al. 1989), this manuscript is certainly an important contribution to the development, validation and application of wind sensing using molecular backscattering in recent years. Thus it is of high relevance for the AMT audience, and I recommend publication after revision. The manuscript is clearly structured and written, and contains recent wind measurement examples from 2019 up to 75 km including one of the first validation results for the space-borne wind-lidar on Aeolus.

**General comments**

1) The manuscript lacks some details on the upgrade and development of the lidar and retrieval algorithms (including calibration) during the recent years since 2012 (after the last publications from this group by Souprayen et al. 1999). E.g. please state clearly, which part of the instrument design in ch. 2.2. is new and also provide more details of the upgrade. Please be more specific on instrument details (see also my specific comments) to this part, e.g. in ch. 3.1 it is stated that FPI plates were reconditioned, but not further explained.

I would also recommend providing more details on the calibration (L79-L83), as this is essential for wind retrieval and wind bias. E.g. is the spectral tuning of the FPI only used for monitoring, or is it used during the wind retrieval (as mentioned in L125). If yes how are these functions used (measured, fitted), and used for wind retrieval from the actual measurements of the same day. Also a short description on how calibration constant C in equation (1) is obtained is missing.

2) I have two comments to the statistical comparison approach. I am wondering about a justification of using weighted distances for deriving bias and standard deviation in Ch. 3. I would like to see a clear justification of this approach, because I consider this as unusual for instrument inter-comparisons, and provide a short description (e.g. equation), how this was implemented. But overall I would recommend deriving these statistical numbers on bias/std. with and without this weighted approach.

My second comment here is related to a missing statistical comparison of the horizontal wind speed (from u and v-components, and possibly wind direction). I would propose to add this quantity to chapter 3, and specifically provide a scatterplot (as Fig. 4d) and statistical numbers (as part of Table 1). I would also propose to add the statistics of all radiosonde comparisons to Table 1 as an additional row, and discuss these numbers in the text.

**Specific comments**

I have listed a number of specific and minor comments below.

| Line | Comment |
| --- | --- |
| 12, 523 | Provide numbers for vertical and temporal resolution; "high resolution" is different for several application areas |
| 29 | Provide a reference for deriving wind speed on regular bases from space-borne temperature measurements using geostrophic assumptions. |
| 38 | I would propose to add some more references in the introduction of wind lidars using molecular backscattering, especially here also mention ALADIN and its airborne demonstrator. |
| 58 | Add 1-2 references for Aeolus here. |
| 70 | Parameters of the FPI are introduced here, while the operating wavelength is not stated (at this place of the manuscript). |
| 91, eq 2 | The introduction of parameter $P(z, 40°)$ is missing in the text. |
| 95-99 | The vertical pointing beam is used to compensate for laser frequency drifts, with a value, which is constant for each altitude (average over 15-25 km). Please discuss, if there are or not altitude dependent effects in the calibration, which need to be compensated. |
| 100ff | Please provide more instrumental details, as laser frequency stability (shot-to-shot), laser divergence (at output of beam expander) and laser linewidth. Also FOV of telescope should be provided, as well as diameter of multimode fiber. The method of mode scrambling should be shortly introduced. Also the "reconditioning" of the FPI plates (as mentioned in ch. 3.1) should be explained here (new coating? New polishing?) |
| 150 | Is this equation of the error in units of m/s? Is C the same constant as introduced in eq. (1)? |
| 200 | Figure 3: black circles are hardly visible, e.g. use different colour. |
| 220 | Do you provide numbers for correlation coefficient as $r$ or $r^2$. Please state explicitly in the text and in Table 1. |
| 212 | Please explain the rationale to compute the comparison statistics, by "weighting" the difference with the horizontal offset between the measurements. I think this is very unusual. I would propose to provide statistics without weighting, or at least show both the non-weighted or weighted results. The weighting should be shortly explained (e.g. via an equation). |
| 240 | Figure 4: y-intercept also in units of m/s |
| 314-316 | Please explain, how a possible Mie-induced bias would be recognized in the profiles, e.g. too high or too low values? Do you correct for the Mie-induced bias in the wind retrieval (or any QC), or is it only compensated by the FPI spectral configuration (spacing, FWHM)? |
| 505 | Fig caption 10; provide date of comparison and mean distance of Aeolus observations to OHP; it would be also good to include Aeolus track in Fig. 3 |
| 515 | Please provide distance for altitudes below 5 km of OHP and Aeolus track for spatial variability. Causes could be also related to preliminary nature of Aeolus observations. Have you checked error estimates within Aeolus data products, and |

| | |
|---|---|
| | potentially exclude data with too high errors (e.g. 8-10 m/s)? Have you checked presence of aerosol or cloud layers, which might influence Aeolus Rayleigh wind retrieval? |
| 526 | Please state that this number of 6 m/s refers to random error. |
| 553 | Could you be more specific, how this finding should be considered for spatial and temporal collocation requirement for performing comparisons for space-borne wind lidars as Aeolus. |
| 527 | Please specify which optics could be replaced to improve performance. |
| 533 | The std of 2.2. m/s refers only to 1 component and not the horizontal wind speed. This should be clarified. I would also propose to add statistics for the horizontal wind speed in the conclusion (see my general comment 2). |
| 605, Table 1 | Please add in Table caption if you use R or R=r2 as correlation coefficient; I would also propose to add at least columns for mean difference and standard deviation for horizontal wind speed (squared sum of u,v; and possibly wind direction) and also another row with mean quantities over all days of comparison. |

**References:**

I would propose to add a few more references related to Aeolus (ESA 2008, Stoffelen et al. 2005) and its actual performance (Kanitz et al. 2019, Reitebuch et al. 2019).

ESA (2008): ADM-Aeolus Science Report. ESA SP-1311, ISBN 978-92-9221-404-3, 121 pages.

T. Kanitz, J. Lochard, J. Marshall, P. McGoldrick, O. Lecrenier, P. Bravetti, O. Reitebuch, M. Rennie, D. Wernham, and A. Elfving, "Aeolus First Light – First Glimpse," Proceedings of SPIE 11180, 111801R (2019).

Reitebuch, O., Lemmerz, C., Lux, O., Marksteiner, U., Rahm, S., Weiler, F., Witschas, B., Meringer, M., Schmidt, K., Huber, D., Nikolaus, I., Geiss, A., Dabas, A., Flament, T., Stieglitz, H., Isaksen, L., Rennie, M., de Kloe, J., Marseille, G.-J., Stoffelen, A., Wernham, D., Kanitz, T., Straume, A. G., Fehr, T., von Bismarck, J., Floberghagen, R., and Parrinello, T.: Initial Assessment of the Performance of the First Wind Lidar in Space on Aeolus, Proc. 29th International Laser-Radar Conference, Hefei, China, June 24-28, 2019.

Stoffelen, A., J. Pailleux, E. Källén, J. M. Vaughan, L. Isaksen, P. Flamant, W. Wergen, E. Andersson, H. Schyberg, A. Culoma, R. Meynart, M. Endemann, P. Ingmann (2005): The Atmospheric Dynamics Mission for Global Wind Field Measurements. Bull. Am. Meteor. Soc. 86, 73-87.

**Editorials**

I have recognized the following limited number of editorial

| Line | Comment |
|------|---------|
| 43 | acronyms LiWind and LiOvent should be introduced |
| 70 | 40° is stated here, while L91 and eq. 2 say 41° |
| 73 | FPI instead of FMI |
| 81 | "30 °C" |
| 218 | "For both wind components" |
| 122 | "1-minute" |
| 250 | "The 1:1 line is shown" |
| 304 | acronym LTA |
| 356 | acronym LiO3S |
| 485 | "Aeolus Level 2B" |
| 479 | "dusk/dawn" |
| 669 | format of reference |

---

## Referee Comment (RC2) · Anonymous Referee #2 · 5 Dec 2019

Overall, this is an interesting and well-written paper on one of the very few existing stratospheric / mesospheric wind lidars. Such systems are of interest to the middle atmosphere scientific community, and the paper is well-suited for publication in AMT. I have only a few minor suggestions.

Line 12: It would be good to get some numbers for the improvement, e.g. from typically xx m/s uncertainty to yy m/s uncertainty. Or were the improvements just technical - then also indicate what has improved.

Line 15: Instead of "The evaluation" I suggest to write "An initial evaluation". The present paper, in my opinion, does not provide a full and comprehensive evaluation.

[Figure]

With only a few radiosondes and ECWMF profiles, the statistics are not very compre-
hensive yet.

Line 39: drop "there exists"; "with" → "have"; "which" → "and";

Line 56: drop "preparation of"

Section 2, 2.1: I think it is necessary to give a bit more background on the wind-lidar
measurement principle. I strongly suggest to add a schematic Figure showing the two
(A and B) Fabry-Perot band-passes spectral shapes, as well as the spectral shape of
the backscattered Rayleigh and Mie radiation. Also explain that a Doppler shift of the
return signal will enhance one channel (A or B) while reducing the other (B or A). How
is spectral calibration obtained? I assume by de-tuning the laser with a wavemeter, and
observing the zenith pointing return channel. Please also explain.

Around line 70: Please give the manufacturer of the Fabry-Perot interferometer.

Around line 151: You might want to say here that the uncertainty scales with
$1/\sqrt{t_{acquisition}}$ and / or with $1/\sqrt{\Delta z}$, where $\Delta z$ is the vertical resolution chosen for
data processing.

Fig. 2a: It would be good to show both the $N_A$ and $N_B$ profiles (or their difference if
they are very similar). Maybe also show a raw $R(\theta, z)$ profile?

Line 172: By "noise level" you probably mean the "background noise level"? If yes,
change text. I would assume that the total noise level would increase a bit at lower
altitudes, e.g. at the altitude where low and high gain channels are spliced together.

Around line 177: 12 Comparisons over a 4-year period are not a lot. Please add some
statement why only so few RS comparisons are made, especially since nearby Nimes
launches one or two radiosondes every day.

Line 218: "The both" → "both"

Around line 254: By eye, Fig. 4a and 4b seem to indicate increasing standard deviation

from about 10 to 30 km. How do standard deviation profiles compare to the estimated uncertainty profile from shot noise? Always a good idea to check such estimates. Maybe this warrants an additional Figure?

Section 3.1: Please add some explanation, that the very narrow Mie line alters the spectral shape of the return signal, and that this might affect/ alter the calibration function in Section 2.1.

Figure 6: I suggest that the authors be more critical here. The largest differences between RS wind and Doppler lidar wind do occur near 12 and 17 km, very close to the aerosol / cirrus layers. I don't think the authors should ignore that and simply claim no effect.

Could the Mie effect be reduced / quantified by wavelength scanning the zenith return signal in the presence of aerosol layers, and assume negligible vertical wind?

Line 366: But ECWMF also assimilates stratospheric and mesospheric radiance measurements from satellites, providing a large amount of information on the temperature fields. Since the atmosphere is close to a geostrophic state in the stratosphere and mesosphere, it is not surprising to me that ECWMF winds are quite realistic up to 60 or 70 km.

lines 386/386: Is it the mirrors, or is it the darker sky in the North? Should "due to a better condition ... mirrors of" be replaced by "due to the darker sky seen by "?

Line 534: I am not sure if you have really demonstrated that results are "insensitive" to aerosol. I think "not very sensitive" would be a better statement.

Around line 535: Can you not measure the temperature profile as well (using the Chanin Hauchecorne method)?

Line 538: I don't think the authors have provided "insight". They only showed "examples" . Replace the word?

---

## Author Comment (AC1) · 31 Jan 2020

**Response to Anonymous Referee #1.**

We thank the Anonymous reviewer #1 for a thorough review and useful suggestions, which have been carefully implemented in the revised manuscript. The detailed answers to all comments are provided below.

*General comments*

*1) The manuscript lacks some details on the upgrade and development of the lidar and retrieval algorithms (including calibration) during the recent years since 2012 (after the last publications from this group by Souprayen et al. 1999). E.g. please state clearly, which part of the instrument design in ch. 2.2. is new and also provide more details of the upgrade. Please be more specific on instrument details (see also my specific comments) to this part, e.g. in ch. 3.1 it is stated that FPI plates were reconditioned, but not further explained. I would also recommend providing more details on the calibration (L79-L83), as this is essential for wind retrieval and wind bias. E.g. is the spectral tuning of the FPI only used for monitoring, or is it used during the wind retrieval (as mentioned in L125). If yes how are these functions used (measured, fitted), and used for wind retrieval from the actual measurements of the same day. Also a short description on how calibration constant C in equation (1) is obtained is missing.*

The Section 2 regarding the instrument design, measurement principle and instrument calibration has been entirely reworked, please see the revised text and the answers to specific comments below.

*2) I have two comments to the statistical comparison approach. I am wondering about a justification of using weighted distances for deriving bias and standard deviation in Ch. 3. I would like to see a clear justification of this approach, because I consider this as unusual for instrument intercomparisons, and provide a short description (e.g. equation), how this was implemented. But overall I would recommend deriving these statistical numbers on bias/std. with and without this weighted approach.*

As a matter of fact, the comparison statistics figures in Table 1. are obtained without the horizontal offset weighting. Besides, as mentioned in Sect. 3, the weighting only affects the average standard deviation, whereas the average bias and correlation coefficient are not affected. Please see the answer to specific comment below.

*My second comment here is related to a missing statistical comparison of the horizontal wind speed (from u and v-components, and possibly wind direction). I would propose to add this quantity to chapter 3, and specifically provide a scatterplot (as Fig. 4d) and statistical numbers (as part of Table 1). I would also propose to add the statistics of all radiosonde comparisons to Table 1 as an additional row, and discuss these numbers in the text.*

The results and discussion of statistical comparison for the total wind and wind direction have been added into Sect. 3 and Table 1.

*Specific comments*

*Lines 12, 523 Provide numbers for vertical and temporal resolution; "high resolution" is different for several application areas*

The respective fragment in the abstract and the summary have been modified: "**After a recent upgrade, the instrument gained the capacity of wind profiling between 5 and 75 km altitude with vertical resolution up to 115 m and temporal resolution up to 5 minutes.**"

*Line 29 Provide a reference for deriving wind speed on regular bases from space-borne temperature measurements using geostrophic assumptions.*

Reference provided: **Oberheide J, Lehmacher G A, Offermann D, Grossmann K U, Manson A H, Meek C E, Schmidlin F J, Singer W, Hoffmann P. Vincent R A. Geostrophic wind fields in the stratosphere and mesosphere from satellite data. J. Geophys. Res. 107(D23):8175. doi: 10.1029/2001JD000655, 2002.**

*Line 38 I would propose to add some more references in the introduction of wind lidars using molecular backscattering, especially here also mention ALADIN and its airborne demonstrator*

The following text has been added: "**The direct-detection technique for wind profiling has been successfully realized in an airborne Doppler lidar – A2D, Aeolus Airborne Demonstrator (Reitebuch et al., 2009). A2D instrument served a prototype for the most ambitions endeavor in the context of lidar wind profiling – the first ever satellite-borne Doppler lidar ALADIN (Atmospheric Laser Doppler INstrument) (ESA, 2008; Stoffelen et al., 2005), that has been successfully launched by European Space Agency (ESA) in August 2018 (Kanitz et al., 2019).**"

*Line 58 Add 1-2 references for Aeolus here*
Done

*Line 70 Parameters of the FPI are introduced here, while the operating wavelength is not stated (at this place of the manuscript).*
*Line 91, eq 2 The introduction of parameter P(z, 40°) is missing in the text. 95-99 The vertical pointing beam is used to compensate for laser frequency drifts, with a value, which is constant for each altitude (average over 15-25 km). Please discuss, if there are or not altitude dependent effects in the calibration, which need to be compensated.*
*Line 100ff Please provide more instrumental details, as laser frequency stability (shot-to-shot), laser divergence (at output of beam expander) and laser linewidth. Also FOV of telescope should be provided, as well as diameter of multimode fiber. The method of mode scrambling should be shortly introduced. Also the "reconditioning" of the FPI plates (as mentioned in ch. 3.1) should be explained here (new coating? New polishing?)*

All of the above comments have been carefully implemented in the Sect. 2, please see the revised text. The reconditioning of the OHP wind lidar FPI has not actually been carried out. It was not necessary since the auxiliary experiments have shown that the spectral characteristics of

the FPI have remained unchanged. The reconditioning of FPI has only been done for the La Reunion wind lidar.

*Line 150 Is this equation of the error in units of m/s? Is C the same constant as introduced in eq. (1)?*
This equation describes the error in the response profile *R*, which is unitless. *C* is the same constant as in eq. (1). This has been clarified in the text.

*Line 200 Figure 3: black circles are hardly visible, e.g. use different colour*
The colour of the circles in Fig. 3 (now Fig. 4) has been changed.

*Line 220 Do you provide numbers for correlation coefficient as r or r^2. Please state explicitly in the text and in Table 1.*
We provide the Pearson's correlation coefficient *r*. This has been specified in the text and in the Table 1 caption.

*212 Please explain the rationale to compute the comparison statistics, by "weighting" the difference with the horizontal offset between the measurements. I think this is very unusual. I would propose to provide statistics without weighting, or at least show both the non-weighted or weighted results. The weighting should be shortly explained (e.g. via an equation).*
We have used the horizontal offset weighting for computing the averages of comparison statistics in order to evaluate the effect of the spatial variability of the horizontal wind components. The results of intercomparison in Sect. 3 and Table 1 are computed as ordinary arithmetic averages. The following text and equation have been added:
**"For evaluating the effect of the horizontal offset between the lidar and RS measurements we computed the offset-weighted averages of the intercomparison statistics and compared them with the ordinary averages. The weight for each individual value is defined as $w = 1 - \overline{D}/D\_max$, where $\overline{D}$ is the mean distance between the lidar and RS sampling locations and $D\_max$ is the maximum distance amounting to 69 km (Table 1). We note that the horizontal-offset weighting of the differences neither affects the mean difference nor the mean correlation but reduces the standard deviation for the wind components and total wind by about 0.2 m/s."**

*Line 240 Figure 4: y-intercept also in units of m/s*
The figures' legend has been modified accordingly

*314-316 Please explain, how a possible Mie-induced bias would be recognized in the profiles, e.g. too high or too low values? Do you correct for the Mie-induced bias in the wind retrieval (or any QC), or is it only compensated by the FPI spectral configuration (spacing, FWHM)?*
The Mie-induced bias would appear as sharp enhancement in the wind profile towards higher absolute values. Such a bias may appear in case of the spectral detuning of the FPI bandpasses with respect to the laser backscattered line. The Mie bias can be corrected for, however in reality this is required only in the case of cirrus clouds with scattering ratio above 20 or so. Otherwise, the correction is unnecessary as we demonstrate in the article.
The following paragraph has been added in the beginning of Sect. 3.1:

**"Although the Mie-backscattered line is narrow (0.08 pm) compared to the thermally-broadened Rayleigh line (2 – 2.4 pm) the intensity of the former may be substantially higher and thereby alter the spectral shape of the return signal. In this case, a disproportionally larger flux would be transmitted through one of the FPI bandpasses, affecting its calibration function and introducing a bias into the wind retrieval within the particle layer. The sensitivity to Mie scattering can be reduced by increasing the FPI spectral spacing, however this also reduces the sensitivity to the Doppler shift. The optimal spectral configuration of the FPI has been established on the base of a theoretical model carried out by Souprayen et al. (1999b). They found that for observable stratospheric wind velocities, the residual Mie-induced error is less than 1 m/s for the scattering ratio $R$=10, which is characteristic of a cirrus cloud readily visible to an unaided eye."**

*Line 505 Fig caption 10; provide date of comparison and mean distance of Aeolus observations to OHP; it would be also good to include Aeolus track in Fig. 3*

Date and mean distance have been added to the figure caption. The Aeolus track has been added into Fig. 3 (now Fig. 4).

*Line 515 Please provide distance for altitudes below 5 km of OHP and Aeolus track for spatial variability. Causes could be also related to preliminary nature of Aeolus observations. Have you checked error estimates within Aeolus data products, and potentially exclude data with too high errors (e.g. 8-10 m/s)? Have you checked presence of aerosol or cloud layers, which might influence Aeolus Rayleigh wind retrieval?*

The distance between RS and Aeolus (91 km) has been provided in the text. The preliminary nature of the Aeolus data is clearly articulated in the text and we refrained from a further discussion on the ALADIN data quality. Indeed, the variability of lower-tropospheric winds on a scale of 100 km in the vicinity of Alps can be much larger than the measurement errors. The Rayleigh-clear profiles that were used for this particular validation case did not feature any error anomalies. From the ground-based lidar and AERONET measurements, we noted an absence of clouds and tropospheric aerosol layers during the period of measurements.

*526 Please state that this number of 6 m/s refers to random error.*

Done.

*553 Could you be more specific, how this finding should be considered for spatial and temporal collocation requirement for performing comparisons for space-borne wind lidars as Aeolus*

The respective sentence has been modified: **"This finding is to be considered for Aeolus wind validation activities in a sense that a precise temporal collocation may be more important than the spatial collocation of the measurements."**

*527 Please specify which optics could be replaced to improve performance*

The respective sentence has been modified: **"We note that the vertical range can potentially be extended to 3 – 80 km through replacement of the beam-commuting and beam-splitting mirrors, for which the resources are available."**

*533 The std of 2.2. m/s refers only to 1 component and not the horizontal wind speed. This should be clarified. I would also propose to add statistics for the horizontal wind speed in the conclusion (see my general comment 2).*

We have provided the statistical figures for the total wind speed and direction in the summary section.

*605, Table 1 Please add in Table caption if you use R or R=r2 as correlation coefficient; I would also propose to add at least columns for mean difference and standard deviation for horizontal wind speed (squared sum of u,v; and possibly wind direction) and also another row with mean quantities over all days of comparison.*

All done.

*References: I would propose to add a few more references related to Aeolus (ESA 2008, Stoffelen et al. 2005) and its actual performance (Kanitz et al. 2019, Reitebuch et al. 2019).*

All the suggested references have been added into the introduction and Sect. 5.

***Editorials***
All done.

---

## Author Comment (AC2)

**Response to Anonymous Referee #2.**

We thank the Anonymous reviewer #2 for a positive review and useful comments. We provide below the detailed answers to them.

Line 12: It would be good to get some numbers for the improvement, e.g. from typically xx m/s uncertainty to yy m/s uncertainty. Or were the improvements just technical - then also indicate what has improved.

The abstract has been modified to clarify the improvement: "A direct-detection Rayleigh-Mie Doppler lidar for measuring horizontal wind speed in the middle atmosphere (10 to 50 km altitude has been deployed at Observatoire de Haute Provence (OHP) in southern France since 1993. After a recent upgrade, the instrument gained the capacity of wind profiling between 5 and 75 km altitude with vertical resolution up to 115 m and temporal resolution up to 5 minutes."

*Line 15: Instead of "The evaluation" I suggest to write "An initial evaluation". The present paper, in my opinion, does not provide a full and comprehensive evaluation. With only a few radiosondes and ECWMF profiles, the statistics are not very comprehensive yet.*

We believe that 12 spatiotemporally-collocated radiosoundings conducted in various atmospheric conditions over a period of 4 years is sufficient for the instrument performance evaluation. Note that all the intercomparison measurements have been conducted in a "campaign" regime, which is resourceful and costly.

*Line 39: drop "there exists"; "with"*  $\rightarrow$  *"have"; "which"*  $\rightarrow$  *"and";*

Line 56: drop "preparation of"

Line 218: "The both"  $\rightarrow$  "both"

All done.

Section 2, 2.1: I think it is necessary to give a bit more background on the wind-lidar measurement principle. I strongly suggest to add a schematic Figure showing the two (A and B) Fabry-Perot band-passes spectral shapes, as well as the spectral shape of the backscattered Rayleigh and Mie radiation. Also explain that a Doppler shift of the return signal will enhance one channel (A or B) while reducing the other (B or A). How is spectral calibration obtained? I assume by de-tuning the laser with a wavemeter, and observing the zenith pointing return channel. Please also explain.

The Section 2 regarding the instrument design, measurement principle and instrument calibration has been entirely reworked, please see the revised text. A figure showing the spectral shapes of backscattered line and FPI bandpasses has been added.

Around line 70: Please give the manufacturer of the Fabry-Perot interferometer.

It is StigmaOptique, a small French company that does not exist anymore, we mentioned the name in the text.

Around line 151: You might want to say here that the uncertainty scales with  $1/\sqrt{tacquisition}$  and /or with  $1/\sqrt{\Delta z}$ , where  $\Delta z$  is the vertical resolution chosen for data processing.

Thank you for the suggestion. Done.

Fig. 2a: It would be good to show both the NA and NB profiles (or their difference if they are very similar). Maybe also show a raw  $R(\vartheta, z)$  profile?

The NA and NB profiles look identical on such a plot, whereas the difference between them in MHz doesn't have much physical meaning. Meanwhile, the response profile R is not much different from the horizontal wind profile as the latter is obtained by multiplying R by the instrumental constant, which only slightly varies with temperature.

Line 172: By "noise level" you probably mean the "background noise level"? If yes, change text. I would assume that the total noise level would increase a bit at lower altitudes, e.g. at the altitude where low and high gain channels are spliced together.

Yes, we mean the background noise level. The splicing of low and high gain signals is done where the former is orders of magnitude above the background level.

Around line 177: 12 Comparisons over a 4-year period are not a lot. Please add some statement why only so few RS comparisons are made, especially since nearby Nimes launches one or two radiosondes every day.

The wind lidar validation experiment was conceived to rely exclusively on the reference measurements by GPS radiosondes collocated with the lidar acquisition in time and, as close as possible, in space. The Nimes radiosoundings are too far away (>100 km) and not always collocated in time, which would make the attribution of the discrepancies in the wind profiles ambiguous.

Around line 254: By eye, Fig. 4a and 4b seem to indicate increasing standard deviation from about 10 to 30 km. How do standard deviation profiles compare to the estimated uncertainty profile from shot noise? Always a good idea to check such estimates. Maybe this warrants an additional Figure?

This aspect is discussed in the original version of the article around line 255. We have included the statistical error profile in Fig. 4 (now fig. 5).

Section 3.1: Please add some explanation, that the very narrow Mie line alters the spectral shape of the return signal, and that this might affect/ alter the calibration function in Section 2.1.

The following paragraph has been added in the beginning of Sect. 3.1:

"Although the Mie-backscattered line is narrow (0.08 pm) compared to the thermallybroadened Rayleigh line (2 - 2.4 pm) the intensity of the former may be substantially higher and thereby alter the spectral shape of the return signal. In this case, a disproportionally larger flux would be transmitted through one of the FPI bandpasses, affecting its calibration function and introducing a bias into the wind retrieval within the particle layer. The sensitivity to Mie scattering can be reduced by increasing the FPI spectral spacing, however this also reduces the sensitivity to the Doppler shift. The optimal spectral configuration of the FPI has been established on the base of a theoretical model carried out by Souprayen et al. (1999b). They found that for observable stratospheric wind velocities, the residual Mie-induced error is less than 1 m/s for the scattering ratio R=10, which is characteristic of a cirrus cloud readily visible to an unaided eye."

Figure 6: I suggest that the authors be more critical here. The largest differences between RS wind and Doppler lidar wind do occur near 12 and 17 km, very close to the aerosol / cirrus layers. I don't think the authors should ignore that and simply claim no effect. Could the Mie effect be reduced / quantified by wavelength scanning the zenith return signal in the presence of aerosol layers, and assume negligible vertical wind?

The Mie-induced bias would appear as sharp enhancement in the wind profile towards higher absolute values. It would be closely correlated with the scattering ratio, which is obviously not the case here. Such a bias may appear in case of the spectral detuning of the FPI bandpasses with respect to the laser backscattered line. The Mie bias can be corrected for, however in reality this is required only in the case of cirrus clouds with scattering ratio above 20 or so. Otherwise, the correction is unnecessary as we demonstrate in the article.

Line 366: But ECWMF also assimilates stratospheric and mesospheric radiance measurements from satellites, providing a large amount of information on the temperature fields. Since the atmosphere is close to a geostrophic state in the stratosphere and mesosphere, it is not surprising to me that ECWMF winds are quite realistic up to 60 or 70 km.

To the best of our knowledge, there are no operational radiance measurements in the mesosphere that are assimilated into ECMWF model. The highest channel of AMSU (ch14) peaks at around 43 km.

lines 386/386: Is it the mirrors, or is it the darker sky in the North? Should "due to a better condition . . . mirrors of" be replaced by "due to the darker sky seen by "?

It is the mirrors and the alignment issues that lower the signal strength for the East line-of-sight. The sky background is the same for both directions

*Line 534: I am not sure if you have really demonstrated that results are "insensitive" to aerosol. I think "not very sensitive" would be a better statement.*

A statement "not very sensitive" would have to be quantified, whereas this is not possible as we did not see any measureable effect.

**Around line 535: Can you not measure the temperature profile as well (using the Chanin Hauchecorne method)?**

To measure the temperature profile using the Chanin Hauchecorne method it is necessary to acquire the full spectrum of the backscattered signal proportional to the atmospheric density. In the case of our Doppler lidar the signal is convoluted with the spectral transmission of the Fabry-

Pérot interferometer. The temperature retrieval would thus be prone to a much larger error than using the dedicated LTA lidar instrument at OHP.

*Line 538: I don't think the authors have provided "insight". They only showed "examples" . Replace the word?*

Done.

---

## Author Response (AR2)

**Response to the remarks by Referee #1**

We thank the anonymous referee for attention to the manuscript. All the remarks have been addressed in the revised text. The following sentence has been added regarding the FPI calibration: "
[revised manuscript text omitted]

775